# Neutral, Negative, or Negligible? Changes in Patient Perceptions of Disease Risk Following Receipt of a Negative Genomic Screening Result

**DOI:** 10.3390/jpm10020024

**Published:** 2020-04-17

**Authors:** Kelsey Stuttgen, Joel Pacyna, Iftikhar Kullo, Richard Sharp

**Affiliations:** 1Biomedical Ethics Research Program, Mayo Clinic, Rochester, MN 55901, USA; stuttgen.kelsey@mayo.edu (K.S.); pacyna.joel@mayo.edu (J.P.); 2Department of Health Sciences Research, Mayo Clinic, Rochester, MN 55901, USA; 3Department of Cardiovascular Medicine, Mayo Clinic, Rochester, MN 55901, USA; kullo.iftikhar@mayo.edu

**Keywords:** genomic screening, return of results, negative results, risk perception

## Abstract

Most individuals who undergo genomic screening will receive negative results or results not sufficient to warrant a clinical response. Even though a majority of individuals receive negative results, little is known about how negative results may impact individuals’ perception of disease risk. Changes in risk perception (specifically reductions in perceived risk) may affect both probands and their family members if inaccurate information is communicated to family members. We surveyed patients who received negative results as part of their participation in a genomic screening study and assessed their perceptions of disease risk following receipt of results. Participants had either hyperlipidemia or colon polyps (or both) and received their negative genomic screening results by mail. Of 1712 total individuals recruited, 1442 completed the survey (84.2% completion rate). Approximately one quarter of individuals believed their risk for heart disease to be lower and approximately one third of individuals believed their risk for colon cancer to be lower after receiving and evaluating their negative genomic screening result. 78% of those who believed their risk for one or both diseases had declined had already shared or intended to share their result with family members. Our study suggests patients may interpret a negative genomic screening result as implying a reduction in their overall disease risk.

## 1. Introduction

As genomic screening is increasingly incorporated into both clinical and research settings [1], it is critical we understand how genomic screening results are interpreted by and affect individuals. While there are data on how individuals interpret positive results [2,3,4,5,6,7,8], there are little data on how individuals interpret negative results.

We use the term “negative” to refer to genomic screening results that are not sufficient to warrant a clinical response. In the context of a genomic screening test, such a result might also be considered “neutral” with respect to the information it provides for risk characterization. Negative results from many genetic tests provide limited information. While a negative result may indicate a person is not affected by a particular disorder, is not a carrier of a specific genetic mutation, or is not at increased risk of developing a certain disease, it is possible that the test missed a disease-causing genetic alteration. Many tests cannot detect all genetic changes that can cause a particular disorder [9], and future research may identify more disease-causing changes. Nonetheless, a negative result can provide clinical insights for individuals who are being evaluated for a pathogenic variant that is known to run in that patient’s family. These subtleties add to the interpretive and communication challenges associated with reporting negative genomic test results.

Despite the lack of data on how individuals interpret negative results, it has been generally assumed by providers that returning negative results to patients is unproblematic. However, there is speculation that negative results may be misinterpreted by individuals and may cause inaccurate risk perceptions, negative psychological impacts, and negative impacts on health behaviors [10,11,12]. In particular, prior studies have explored how genetic determinism may override social determinants of health [13], and it is possible that individuals’ risk perception of a condition or disease may be lowered after receiving negative results, even when individuals are still at risk for a particular disease. This may cause a decline in health behaviors such as healthy diet, exercise, and screening activities such as mammographs and colonoscopies.

Furthermore, the impact of misunderstandings about negative results may be amplified when genetic risk information is inaccurately conveyed to family members. This may cause risk a shift in broader perceptions of disease risk and negative impacts on health behaviors in family members.

These considerations highlight the need to understand how negative results are interpreted by and affect patients, especially given that the number of patients receiving negative genomic screening results is increasing considerably [1]. This study assessed how negative genomic screening results affect individuals’ perceptions of disease risk. The study aimed to determine whether patients had lower disease risk perception after receiving a negative genomic screening result, to describe factors that might be associated with a tendency to downgrade perceived disease risk in light of receiving negative results, and to investigate the effects of such changes on risk perception, including whether participants who lowered disease risk perception shared theirs result with family members. We believe this is the first major study to examine the potential harms that may be associated with the communication of a negative genomic screening result.

## 2. Materials and Methods

### 2.1. Setting and Participants

This study was based on the Return of Actionable Variants Empiric (RAVE) Study, conducted as part of the eMERGE consortium, which was funded by the U.S. National Institutes of Health. Results from genomic sequencing were returned to study participants [14]. Participants in this study were members of the Mayo Clinic Biobank and had hyperlipidemia and/or colon polyps. A study flow diagram, more detailed participant criteria, and a list of genes can be found in Kullo et al. [14]. Though not required for participation, pre-test genetic counseling was available at no cost to participants.

Participants with genetic results indicating a need for medical follow up (i.e., “positive screening results”) received results either in person or via phone by a licensed genetic counselor. Participants with genetic results that did not indicate a need for medical follow up (i.e., “negative screening results”) received their results via postal mail. Participants received results between 21 and 25 months after they consented to the study. Materials included in the mailing included both a one-page letter summarizing test results (see Appendix A) and a copy of the laboratory report, which was also entered into the patient’s electronic health record. The results letter was developed in collaboration with genetic counselors as well as the community advisory board serving the biobank. Variants of uncertain significance were not returned. Genetic counseling services were available to all participants free of charge, and a phone number to access services was provided in the result letter mailed to participants.

### 2.2. Survey

As we have previously reported, the survey was 82-items and composed of items that examined reactions to results, perceived value of results, perceived risk of disease, perceived understanding of results, familiarity with family history, intentions to share results with family members, and overall experiences with receiving genomic screening.

### 2.3. Data Collection

This study was approved by the Mayo Clinic Institutional Review Board (#15-005013). All participants who received negative genomic screening results and had completed a baseline psychosocial questionnaire after enrollment were invited to complete the survey. A baseline questionnaire administered at the time of consent collected demographic information on participants, as well as data on multiple independent variables of interest, including health literacy, healthcare access, financial stability, insurance coverage, and knowledge about genomic sequencing [15].

Participants received surveys via postal mail approximately 14 days after receiving their negative results materials via postal mail. Results were mailed to participants beginning in April 2018, and the first group of surveys were sent in early May 2018. Those who did not respond to the survey received a reminder with an additional opportunity to participate approximately 30 days after receiving the initial survey.

The survey was processed by trained staff at the Mayo Clinic Survey Research Center. Surveys were date-stamped when returned and entered into a document tracking database. Completed surveys were double-entered by data-entry staff, and periodic quality checks were conducted by research staff. Responses on the survey that were unclear to data-entry staff were reviewed by a study team member. Responses that could be reasonably ascertained were updated in the dataset, while ambiguous responses were entered as missing.

### 2.4. Data Analysis

Data were analyzed using JMP Pro 14 (2018 SAS Institute Inc, Cary, NC, USA). Means, standard deviations, and ranges were calculated for continuous variables, and frequencies and percentages were calculated for categorical variables. Bivariate associations were calculated using Chi-Square, Wilcoxon Rank Sum, and Fishers Exact Test, as appropriate. *P*-values of 0.05 or lower were considered statistically significant.

The primary outcomes of interest for the analysis reported here are: (1) demographic characteristics in participants with at-risk phenotypes associated with adjusting risk downwards (2) changes in perceptions about disease risk after receiving negative genomic screening results, (3) perceived primary cause of hypothetical future disease after receiving negative results, and (4) odds of sharing results with family members.

As we have described previously, genetic knowledge, familiarity with study procedures, and difficulty understanding results scores were computed in the following ways: Genetic knowledge scores were computed by summing correct responses to an 11-item measure administered at baseline which was developed and published by another research group [7]. Response options included “True,” “False,” and “Don’t Know”. Illustrative questions include: “Genome sequencing may find variants in a person’s genes that will increase their chance of developing a disease in their lifetime” and “Even if a person has a variant in a gene that affects their risk of a disease, they may not develop that disease.”

Familiarity with study procedures scores were computed by summing correct responses to nine knowledge questions with response options of “True,” “False”, and “I Do Not Know.” Illustrative questions include: “My genetic test results from the RAVE study have been placed in my electronic health record” and “The genetic testing done as part of the RAVE study cannot detect all genetic variants that may eventually be known to cause disease.” Missing data for individual items were scored as “incorrect.”

Difficulty understanding results scores were computed from responses to the following three survey items answered on a 5-point Likert scale of agreement (Strongly Agree, Agree, Neither Agree Nor Disagree, Disagree, Strongly Disagree): (1) “When I first read the letter describing my test results, it was difficult to understand,” (2) “I felt the lab report was difficult to understand,” and (3) “I still have questions about what my genetic test results mean.”

Difficulty understanding results scores were standardized with the familiarity with study procedures scores (range of 0 to 9), we scored agreement (Strongly Agree/Agree) to each perception question as 0, neutral responses (Neither Agree Nor Disagree) were coded as 1.5, and disagreement (Disagree/Strongly Disagree) was coded as 3. Recoded scores for each question were summed, resulting in a single score of difficulty understanding results ranging from 0 to 9.

Changes in risk perceptions about disease risk after receiving negative genomic screening results were assessed by responses to the following four survey items in the follow -up survey: “Before my genetic test, I would have said that my risk of colon cancer is…,” “After my genetic test, I would have said that my risk of colon cancer is…,” “Before my genetic test, I would have said that my risk of heart disease is…,” and “After my genetic test, I would have said that my risk of heart disease is….” All four survey items had response choices “higher than the general population,” “the same as the general population,” and “lower than the general population.”

Perceived primary cause of hypothetical future disease was assessed by responses to the survey items: “If I were to get colon cancer, it would be the result of my lifestyle choices and diet, not my genes” and “If I were to get colon cancer, it would be the result of my lifestyle choices and diet, not genes.” Both survey items were answered on a 5-point Likert scale of agreement (Strongly Agree, Agree, Neither Agree Nor Disagree, Disagree, Strongly Disagree). Responses were dichotomized by Strongly Agree and Agree versus Neither Agree Not Disagree, Disagree, and Strongly Disagree.

Sharing intentions and behaviors were assessed by the following survey item: “Have you shared your genetic test results with any of the following people in your family (please mark all that apply)? If your family does not include one of the types of family members listed below, or if that person is no longer living, please check ‘Not Applicable.’” The following family members were listed below this question: my spouse or partner, my father, my mother, at least one of my brothers, at least one of my sisters, at least one of my adult sons, at least one of my adult daughters. Each family member listed had the following response options: “yes,” “no,” “no, but I plan to,” and “not applicable.”

## 3. Results

There were 5110 participants who met eligibility criteria for our study and were invited to participate. Of those, 2538 responded to the study invitation, consented, and pursued genomic screening [12]. Only eight of the 5110 individuals who were invited to participate elected to have pretest genetic counselling, which was provided free of cost. One hundred and eighteen individuals (4.6%) received a “positive” genomic screening result, 2416 individuals (95.2%) received a negative genomic screening result, and four individuals withdrew from the study. Though genetic counseling services were available to all participants free of charge, only four participants utilized genetic counseling services after receiving a negative result.

Surveys were mailed to 1712 individuals who had previously completed a baseline demographic survey. Of 1712 total individuals, 1442 completed the survey (84.2% completion rate). Demographic characteristics of those who completed the survey and those who adjusted their perceived risk downwards after receiving negative results are summarized in Table 1. Our sample was primarily older (mean age = 60.8), white, (96.5%) individuals, and had more women (57.6%) than men (42.2%).

Shifts in risk perception are shown in Table 2, Figure 1, and Figure 2. The most common response after receiving a negative genomic screening result was that participants did not shift their perceived disease risk. The majority of individuals indicated their risk was “the same as” the general population, which they also indicated before receiving a negative result. However, more individuals had concordant pre- and post-ROR risk perception for heart disease (987 individuals; 71.7%) than for colon cancer (931 individuals; 67.6%).

The second most common response after receiving a negative genomic screening result was that participants lowered their risk perception. A total of 402 (29.2%) individuals reported lower post-ROR risk perception for colon cancer. Of these, 230 individuals had a phenotype (colon polyps) and 172 did not. A total of 354 (25.7%) of individuals reported lower post-ROR risk perception for heart disease. Of these, 255 individuals had a phenotype (hyperlipidemia) and 99 did not.

The least common response after receiving a negative genomic screening result was that participants increased their risk perception. Forty-five (3.3%) individuals reported higher post-ROR risk perception for colon cancer, and of these, 23 had colon polyps and 22 did not. Thirty-six (2.6%) of individuals reported higher post-ROR risk perception for heart disease, and of these, 25 had hyperlipidemia and 11 did not.

A breakdown of perceived risk of colon cancer in participants with and without colon polyps before and after receiving results is shown in Figure 1. A breakdown of perceived risk of heart disease in participants with and without hyperlipidemia before and after receiving results is shown in Figure 2.

Individuals who had colon polyps and hyperlipidemia were evaluated across demographic variables in Table 3 to assess whether certain demographic variables might be associated with adjusting risk downwards. In individuals with colon polyps, no demographic variables were associated with adjusting risk down. In heart disease, however, both sex and familiarity with study procedures were associated with adjusting risk downwards. Females were more likely to adjust risk downward than males. 

In addition to assessing the overall degree of risk assigned, we were interested in assessing where participants localized that risk. Specifically, whether participants felt their negative genomic screening result made it more likely that a future disease diagnosis would be the result of environmental or lifestyle factors and not genetic factors. Participants’ primary attribution of cause of future hypothetical disease diagnoses after receiving negative results were evaluated across demographic variables and are shown in Table 4. Participants with higher levels of education, and those with adequate health literacy, were more likely to believe that a future diagnoses of heart disease would be the result of a lifestyle choice, not genes.

Participants who did and did not adjust their risk downward and shared or intended to share results with family members are shown in Table 5. Most participants (78.7%) who adjusted risk downward for colon cancer after receiving results shared intended to share their results with family members. Similarly, most participants (78.1%) who adjusted risk downward for heart disease after receiving results shared or intended to share their results with family members.

## 4. Discussion

Outside of genomics, the term “screening” in medicine often refers to checking for biomarkers associated with susceptibility to disease or early signs of disease. For example, screening for breast cancer involves mammograms, which check for abnormal masses which may be early signs of cancer or already cancerous. Within genomics, “screening” refers to assessing genetic factors for the risk developing a future disease. Our study aimed to examine the extent to which patients might interpret negative results as indicative of a lower overall risk for developing disease.

Individuals in our sample had a phenotype of hyperlipidemia or colon polyps and negative genomic screening results, which alone does not indicate that an individual is at lower-than-average risk for developing heart disease or colon cancer. In our study we observed that approximately one quarter of individuals lowered their perceived risk for heart disease and approximately one third of individuals lowered their perceived risk for colon cancer after receiving a negative genomic screening result. These data provide the first empirical evidence in support of concerns that patients may be inclined to downgrade their perceived risk in light of receiving a negative result.

Lowered risk perception after receiving negative genomic screening results is worrisome if it results in a change in health behaviors or family communication patterns. Individuals may feel it is less important to follow a healthy diet and exercise plan, see a physician, or undergo recommended health screenings such as colonoscopies, mammograms, PAP smears, etc. [16]. Our data indicate that participants who adjusted risk downward for either colon cancer and heart disease were more likely to have shared or intend to share with family members. Lowered risk perception may cause individuals to feel it is less important to inform their family members of family health risks, or alternatively, may cause individuals to misinform family members about health risks they believe they are at lower risk for (or no longer at risk for) after receiving negative genomic screening results. This potential change in health narratives in families could expand the impact of negative results interpretation beyond individuals.

Another reason why lowered risk perception is worrisome is that if diagnosed with a disease in the future, individuals may attribute the diagnosis purely to their lifestyle choices under the false pretense that their negative genomic screening results eliminated genetic risk for a particular disease. This potentially misattributed cause of disease could have adverse psychological effects.

While there is limited literature on how negative results (especially negative genomic screening results) are interpreted by and impact individuals, available data are concordant with our findings and suggest that individuals may have difficulty interpreting negative results and understanding the limitations associated with negative results. In one study, women at increased risk for breast cancer based on family history who received negative results via current BRCA1/2 testing often misconceived that results either meant nothing, or that their risk for developing breast cancer was as low as that of the average woman [17]. Our results are concordant with these prior studies such that our participants also had difficulty interpreting the limitations associated with negative results and may have a false sense of relief about their disease risk, since some of our participants reported their disease risk was lowered after receiving a negative result.

Given the diagnostic connotations of the term “screening” in areas of medicine outside of genomics, it is possible that individuals in our study interpreted their negative result to be a negative diagnostic finding. Interpretation of negative genomic screening results in a diagnostic context rather than a screening context has been reported in other studies. In one study, hypocholesterolemia patients felt both relieved by their negative genomic results and disappointed that a conclusive answer to the question of the cause of their condition had not been found [18]. These results suggest individuals may interpret genomic screening results in a diagnostic context.

A higher proportion of individuals with a phenotype for colon cancer had lower post-ROR risk perception than individuals with a phenotype for heart disease. It is possible individuals perceive colon cancer to be more strongly associated with genetic risk than heart disease. Family history can determine the age at which individuals begin getting colonoscopies such that individuals with a family history of colon cancer may begin getting colonoscopies at a younger age than the general population [19]. This illustrates a clear connection for patients between genetic factors and cancer risk. While bloodwork may be performed on patients with a family history of heart disease, the purpose of the bloodwork may not to be clear to patients since the burden on the patient is much lower for bloodwork than for colonoscopies. Additionally, conversations between patient and provider may be different after diagnosis of a colon polyp compared to diagnoses of hyperlipidemia. Conversations after diagnoses of hyperlipidemia may be more focused on lifestyle changes in diet and exercise, which influence the risk of heart disease to be more strongly associated with environmental rather than genetic factors.

A higher post-ROR risk perception of colon cancer risk and heart disease risk could also be explained by a change in personal or family history since the time of the baseline survey. Changes in personal or family history might include the presence of colon polyps, bloodwork indicating early signs of heart disease, or a family member diagnosed with either cancer or heart disease. This could also be due to survey error/misinterpretation of the question asked.

There does not appear to be demographic variables consistently associated with participants who adjust risk downward or believe a future hypothetical disease diagnoses would be a result of lifestyle choices. This suggests the return of results letter did not adequately communicate the limitations and implications (or lack thereof) of the negative results to participants. Education and health literacy, for example, were not the primary factors driving the downgrading of risk we observed. Since we were unable to identify subgroups at increased likelihood to downgrade risk, targeted interventions to specific populations or demographics would likely not be the best response to these concerns. Rather, broad educational activities would be more appropriate.

Further research is needed to clarify how negative results affect risk perception and how changes in risk perception result in changes in health behaviors or family communication of genetic risk information. Other studies have observed various changes in health-behaviors after receiving a negative result. In HIV-negative men who have sex with men, behavioral choices following multiple negative test results varied from continuing safe sex practices to increasing risk behaviors [17]. In another study, negative screening tests did not falsely reassure individuals at high risk for developing diabetes, whose intentions for behavioral change were essentially the same for negative results recipients and the control group [20]. A complementary study showed that four years after follow-up, screening negative for diabetes did not lead to long-term changes in lifestyle [21]. Though there are limitations as to the extent of parallels that can be drawn between these studies and genomic screening, these studies are examples of settings that may provide insights into the communication of a negative genomic screening results that may aid future research.

### Limitations

Due to the nature of our study, which involved a pre- and post-results survey, we were not able to assess whether lowered risk perception resulted in changes in health behaviors or family communication patters. Further research is needed to investigate how changes in risk perception affect changes in health behavior and family communication patterns.

Additionally, the nature of our survey questions did not allow us to assess the magnitude of change in risk perception. It is possible that individuals may have had only a slight increase or decrease in risk perception. However, what is clear is the trend of how risk perceptions change after receiving negative genomic screening results. It is not clear what messages are important to convey to individuals in terms of whether and how much their risk has been reduced. In the letter utilized in this study, we did not attempt to speak to how much risk should go up and down. We acknowledge the shift in risk perception observed in this study could be a result of how results were communicated to individuals in the letter used in our study.

Another limitation of our study is that we were not able to assess whether and what other factors may have influenced risk perception during the study period. It is possible participants were adjusting their risk perception in response to a number of factors that this study did not measure. These factors may have included changes in personal or family member’s health status and changes in health behaviors such as cessation of smoking, changes in diet, or changes in exercise habits.

The modality of returning negative genomic screening results in our study may have influenced individuals’ perception of the meaning of the results and in turn, resulted in lowering of risk perception. At the authors’ institution, most often “bad news,” or results indicating disease, are returned to patients in person, whereas “good news,” or results indicating a clean bill of health, are returned by mail, electronic message through a patient portal, or by phone. It is possible receiving results by letter caused individuals to interpret negative genomic screening results as a clean bill of health. Additionally, results in our study were communicated within the context of a research study and not a clinical interaction, which may have influenced individuals’ perception of the meaning of results.

## 5. Conclusions

Our results support concerns that a significant proportion of patients may (wrongly) interpret a negative genomic screening result as implying a reduction in their overall disease risk. We observed this downgrading of disease risk perceptions across the two phenotypes and associated diseases of interest in our study, and as such, this may be a more general problem that cannot be attributed to a specific phenotype, disease, or contributing factor, such as education or health literacy. This is a worrisome issue that should be the subject of additional research.

## Figures and Tables

**Figure 1 jpm-10-00024-f001:**
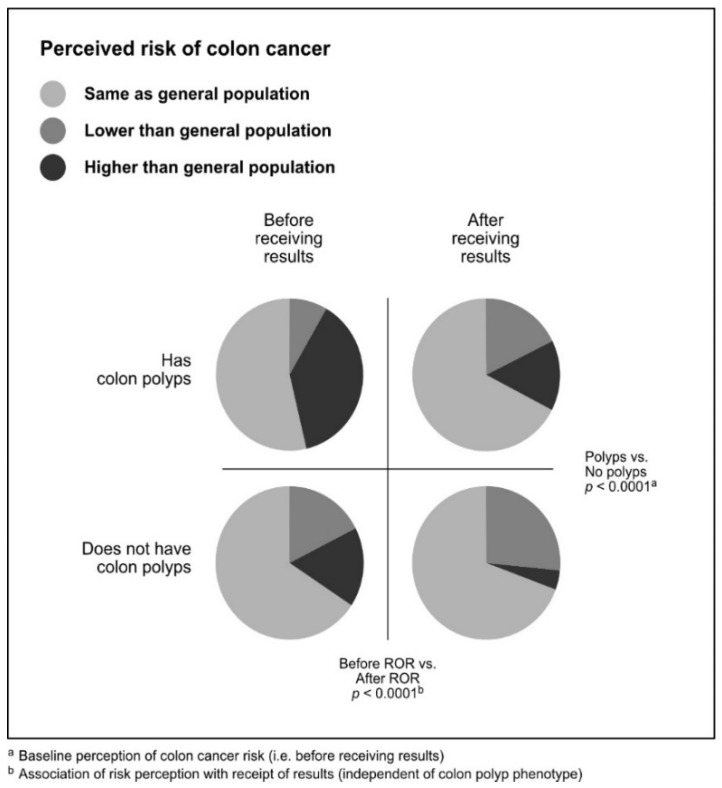
A comparison of perceived risk of colon cancer in patients before and after receiving negative genomic screening results.

**Figure 2 jpm-10-00024-f002:**
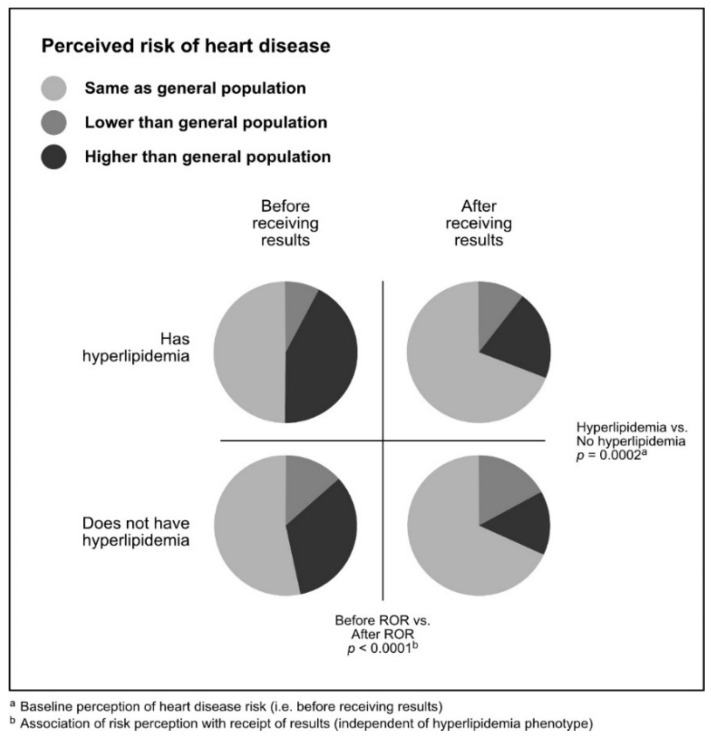
Perceived risk of heart disease in participants with and without hyperlipidemia before and after receiving results.

**Table 1 jpm-10-00024-t001:** Demographic characteristics of participants who pursued genomic screening (*n* = 1442) and those who adjusted their risk down after receiving negative results.

Characteristic	Total N (%)	Adjusted Risk Down N (%)
N	1442	604
Sex		
Male	601 (42.4)	225 (37.8)
Female	817 (57.6)	370 (62.2)
Age (years) at study invitation		
Mean; SD	60.8; 7.3	60.8 (7.4)
Range	27–71	28–71
Race		
White	1392 (96.5)	590 (97.7)
Other	50 (3.5)	14 (2.3)
Ethnicity		
Non-Hispanic	1413 (99.7)	593 (99.8)
Hispanic	4 (0.3)	1 (0.2)
Marital Status		
Married / partnered	1185 (83.6)	500 (84.0)
Not married / partnered	233 (16.4)	95 (16.0)
Genetic Knowledge		
Mean, SD	8.3, 2.2	8.3 (2.1)
Range	0–11	0–11
Education		
Grades 9-11	1 (0.1)	0 (0.0)
Grade 12/GED	167 (11.7)	67 (11.2)
College 1-3 years	502 (35.3)	228 (38.0)
College 4+ years	405 (28.4)	163 (27.2)
Grad/professional school	349 (24.5)	142 (23.7)
Health Literacy		
Adequate	1338 (92.8)	557 (93.0)
Inadequate	104 (7.2)	42 (7.0)
Unable to access physician due to cost	28 (2.0)	11 (1.8)
Financial Situation (income)		
More than enough	1162 (82.3)	483 (81.6)
Just enough	201 (14.2)	89 (15.0)
Have to cut back	40 (2.8)	17 (2.9)
Difficulty paying bills	9 (0.6)	3 (0.5)
Insurance coverage		
None	9 (0.6)	3 (0.5)
Private	1107 (77.8)	465 (77.6)
Public program	307 (21.6)	131 (21.9)
Colon Polyp Phenotype	711 (49.7)	312 (52.1)
Elevated Lipid Phenotype	1011 (70.7)	411 (68.6)

**Table 2 jpm-10-00024-t002:** Perceived risk of colon cancer and heart disease for participant with and without respective phenotypes before and after receiving results as compared to the general population.

	Higher Than General Population	Same as General Population	Lower Than General Population	*p*-Value ^a^
**Perceived risk of colon cancer**				
Has colon polyps				<0.0001
Before receiving results	266 (38.0)	376 (53.7)	58 (8.3)	
After receiving results	104 (14.9)	470 (67.3)	124 (17.8)	
Does not have colon polyps				<0.0001
Before receiving results	119 (16.8)	467 (65.8)	124 (17.5)	
After receiving results	29 (4.1)	492 (69.3)	189 (26.6)	
**Perceived risk of heart disease**				
Has lipids				<0.0001
Before receiving results	425 (42.5)	496 (49.6)	78 (7.8)	
After receiving results	204 (20.5)	688 (69.0)	105 (10.5)	
Does not have high lipids				
Before receiving results	137 (33.3)	220 (53.4)	55 (13.3)	<0.0001
After receiving results	61 (14.8)	280 (68.1)	70 (17.0)	

^a^ Chi-square.

**Table 3 jpm-10-00024-t003:** Participants with at-risk phenotypes who did and did not adjust perceived risk down for colon cancer and heart disease.

	Has Colon Polyps				Has Hyperlipidemia			
		Adjusts Perceived Risk of Colon Cancer Down				Adjusts Perceived Risk of Heart Disease Down		
	All	No	Yes	*p* Value	All	No	Yes	*p* Value
	*n* = 711	*n* = 461	*n* = 237		*n* = 1011	*n* = 734	*n* = 263	
	N (%)	N (%)	N (%)		N (%)	N (%)	N (%)	
Sex				0.17				0.001
Male	326 (46.8)	220 (48.7)	100 (43.1)		424 (42.6)	327 (45.4)	88 (33.8)	
Female	371 (53.2)	232 (51.3)	132 (56.9)		571 (57.4)	394 (54.6)	172 (66.2)	
Age, Mean (SD)	61.6 (6.3)	61.7 (6.3)	61.6 (6.3)	0.99	60.7 (7.6)	60.9 (7.6)	60.2 (7.7)	0.20
Marital status				0.56				0.39
Partnered	586 (84.1)	380 (84.1)	199 (85.8)		835 (83.9)	610 (84.6)	214 (82.3)	
Not partnered	111 (15.9)	72 (15.9)	33 (14.2)		160 (16.1)	111 (15.4)	46 (17.7)	
Education				0.34				0.30
Grades 9-11	1 (0.1)	1 (0.2)	0 (0.0)		1 (0.1)	1 (0.1)	0 (0.0)	
HS Grad/ GED	89 (12.7)	64 (14.1)	24 (10.2)		111 (11.2)	89 (12.3)	20 (7.7)	
College 1-3 y	238 (33.9)	143 (31.5)	89 (37.9)		358 (36.0)	252 (35.0)	99 (37.9)	
College 4+ y	207 (29.5)	138 (30.4)	66 (28.1)		284 (28.5)	203 (28.2)	79 (30.3)	
Grad School	167 (23.8)	108 (23.8)	56 (23.8)		241 (24.2)	176 (24.4)	63 (24.1)	
Race				0.48				0.15
White	691 (97.2)	447 (97.0)	232 (97.9)		973 (96.2)	703 (95.8)	257 (97.7)	
Other	20 (2.8)	14 (3.0)	5 (2.1)		38 (3.8)	31 (4.2)	6 (2.3)	
Health literacy				0.98				0.26
Adequate	661 (94.4)	429 (94.5)	221 (94.4)		932 (93.8)	680 (94.3)	241 (92.3)	
Inadequate	39 (5.6)	25 (5.5)	13 (5.6)		38 (6.2)	41 (5.7)	20 (7.7)	
Self-reported health				0.35				0.13
Excellent	98 (13.8)	57 (12.4)	37 (15.7)		129 (12.8)	100 (13.7)	25 (9.5)	
Very good	328 (46.3)	208 (45.3)	117 (49.6)		492 (48.9)	357 (48.9)	130 (49.4)	
Good	222 (31.4)	151 (32.9)	65 (27.5)		328 (32.6)	226 (31.0)	97 (36.9)	
Fair	55 (7.8)	40 (8.7)	15 (6.4)		55 (5.5)	44 (6.0)	11 (4.2)	
Poor	5 (0.7)	3 (0.7)	2 (0.8)		3 (0.3)	3 (0.4)	0 (0.0)	

**Table 4 jpm-10-00024-t004:** Participant emphasis on negative genomic screening results compared to lifestyle choices in primary attribution of the cause of hypothetical future disease.

	Total	If I Were to Get Colon Cancer, It Would Be the Result of Lifestyle Choices, Not Genes N (%)	*p*-Value	If I Were to Get Heart Disease, It Would Be the Result of Lifestyle Choices, Not Genes N (%)	*p*-Value
Sex			0.62		0.18
Male	60142.4)	261 (42.9)		340 (43.8)	
Female	817 (57.6)	347 (57.1)		436 (371)	
Age, Mean (SD)	60.8 (7.3)	60.7 (7.3)	0.90	60.7 (7.3)	0.74
Marital status			0.85		0.73
Partnered	1185 (83.6)	510 (83.9)		647 (83.4)	
Not partnered	233 (16.4)	98 (16.1)		129 (16.6)	
Education			0.65		0.009
Grades 9-11	1 (0.1)	0 (0.0)		1 (0.1)	
HS Grad/ GED	167 (11.7)	68 (11.1)		84 (10.7)	
College 1-3 y	502 (35.3)	206 (33.7)		251 (32.0)	
College 4+ y	405 (28.4)	183 (30.0)		236 (30.1)	
Grad School	349 (24.5)	154 (25.2)		212 (27.0)	
Race			0.62		0.43
White	1392 (96.5)	595 (93.6)		761 (96.2)	
Other	50 (3.5)	23 (3.7)		30 (3.8)	
Health literacy			0.39		0.03
Adequate	1338 (94.1)	579 (94.8)		746 (95.3)	
Inadequate	84 (5.9)	32 (5.2)		37 (4.7)	
Self-reported health status			0.45		0.26
Excellent	194 (13.5)	75 (12.2)		112 (14.2)	
Very good	686 (47.8)	293 (47.5)		372 (47.3)	
Good	465 (32.4)	210 (34.0)		262 (33.3)	
Fair	85 (5.9)	38 (6.2)		39 (5.0)	
Poor	5 (0.3)	1 (0.2)		2 (0.3)	

**Table 5 jpm-10-00024-t005:** Participants who shared or intended to share results with family members, stratified by participants who adjusted risk down after receipt of results and participants who did not adjust risk down after receipt of results.

	Shared or Intend to Share with Family N (%)	Do not Intend to Share with Family N (%)	*p*-Value
Perceived risk of Colon Cancer after receipt of results			0.002
Adjusted down	325 (78.7)	88 (21.3)	
Did not adjust down	709 (70.6)	296 (29.5)	

Perceived risk of Heart Disease after receipt of results			0.009
Adjusted down	288 (78.1)	81 (22.0)	
Did not adjust down	745 (71.0)	304 (29.0)

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
