# Peer review of "Neutral, Negative, or Negligible? Changes in Patient Perceptions of Disease Risk Following Receipt of a Negative Genomic Screening Result"

_jpm, 2020, doi:10.3390/jpm10020024_

Round 1
Reviewer 1 Report
The paper addresses an important issue and confirms concerns about patient interpretation of negative results that have been raised. It is interesting that very few participants utilized the (free) genetic counselling. An important additional question which is only partially addressed in Tabel S2 is the interpretation of their risk for a different condition due to the negative result ie this with colon polyps, how do they rat etheir risk for heart disease and vice vera. Table S2 is important and should be included in the main document as it is the broader implicatins of the results of additional screening that is done for the ACMG list that is also important.
Please note missing text in Line 45.
Author Response
The paper addresses an important issue and confirms concerns about patient interpretation of negative results that have been raised. It is interesting that very few participants utilized the (free) genetic counselling.
- An important additional question which is only partially addressed in Table S2 is the interpretation of their risk for a different condition due to the negative result (i.e. this with colon polyps, how do they rate either risk for heart disease and vice versa).
Response: Thank you for pointing this out. We agree that this is an important element to include in the analysis. Though we were not able to include this is in Table S2 (which has now become Table 2), we have performed this analysis and placed it in a separate table (Table 3).
- Table S2 is important and should be included in the main document as it is the broader implications of the results of additional screening that is done for the ACMG list that is also important.
Response: Thank you for this helpful suggestion. We have moved Table S2 into the main document and it is now “Table 2.” The other table numbers have been adjusted accordingly.
- Please note missing text in Line 45.
Response: Thank you for catching this embarrassing mistake! We have added the word “unproblematic” to the end of the sentence.

Reviewer 2 Report
This is a well-written presentation of a well-designed study on a topic that we need more evidence about. I have only minor suggestions:
p2, line 45: The last word of the sentence appears to be missing.
p2, line 87: Technically, it would be more grammatically correct to say "the survey was composed of..." (not "comprised of")
Other than that, I would like to see a bit more discussion about the other factors that may have influenced risk perception during the study period, BESIDES the negative test report. We can't ever know what these patients' true risk is, and while we are concerned about changes in risk perception that don't seem warranted, we can't rule out that patients' are actually making nuanced assessments of risk based on a host of other factors this study wasn't able to measure.
Author Response
This is a well-written presentation of a well-designed study on a topic that we need more evidence about. I have only minor suggestions:
- p2, line 45: The last word of the sentence appears to be missing.
Response: Thank you for catching this embarrassing mistake! We have added the word “unproblematic” to the end of the sentence.
- p2, line 87: Technically, it would be more grammatically correct to say "the survey was composed of..." (not "comprised of")
Response: Thank you for pointing this out – we have changed the wording to “composed of.”
- Other than that, I would like to see a bit more discussion about the other factors that may have influenced risk perception during the study period, BESIDES the negative test report. We can't ever know what these patients' true risk is, and while we are concerned about changes in risk perception that don't seem warranted, we can't rule out that patients are actually making nuanced assessments of risk based on a host of other factors this study wasn't able to measure.
Response: We appreciate this suggestion and agree that it is an important element that should be addressed in our manuscript. We have added the following text tot the limitations section (page 14 line 334): “Another limitation of our study is that we were not able to assess whether and what other factors may have influenced risk perception during the study period. It is possible participants were adjusting their risk perception in response to a number of factors that this study did not measure. These factors may have included changes in personal or family member’s health status and changes in health behaviors such as cessation of smoking, changes in diet, or changes in exercise habits.”
